# New Benzofuranoids and Phenylpropanoids from the Mangrove Endophytic Fungus, *Aspergillus* sp. ZJ-68

**DOI:** 10.3390/md17080478

**Published:** 2019-08-18

**Authors:** Runlin Cai, Hongming Jiang, Zhenming Zang, Chunyuan Li, Zhigang She

**Affiliations:** 1School of Chemistry, Sun Yat-sen University, Guangzhou 510275, China; 2College of Materials and Energy, South China Agricultural University, Guangzhou 510642, China; 3South China Sea Bio-Resource Exploitation and Utilization Collaborative Innovation Center, Guangzhou 510006, China

**Keywords:** benzofuranoids, phenylpropanoids, *Aspergillus* sp., *α*-glucosidase, antibacterial activity

## Abstract

Three new benzofuranoids, asperfuranoids A–C (**1**–**3**), two new phenylpropanoid derivatives (**6** and **7**), and nine known analogues (**4**, **5**, and **8**–**14**) were isolated from the liquid substrate fermentation cultures of the mangrove endopytic fungus *Aspergillus* sp. ZJ-68. The structures of the new compounds were determined by extensive spectroscopic data interpretation. The absolute configurations of **1**–**3** were assigned via the combination of Mosher’s method, and experimental and calculated electronic circular dichroism (ECD) data. Compounds **4** and **5** were a pair of enantiomers and their absolute configurations were established for the first time on the basis of their ECD spectra aided with ECD calculations. All isolated compounds (**1**–**14**) were evaluated for their enzyme inhibitory activity against *α*-glucosidase and antibacterial activities against four pathogenic bacteria (*Staphylococcus aureus*, *Escherichia coli*, *Bacillus subtilis*, and *Pseudomonas aeruginosa*). Among them, compound **6** exhibited potent inhibitory activity against *α*-glucosidase in a standard in vitro assay, with an IC_50_ value of 12.4 μM, while compounds **8** and **11** showed activities against *S. aureus*, *E. coli*, and *B. subtilis*, with MIC values in the range of 4.15 to 12.5 μg/mL.

## 1. Introduction

The mangrove forest, growing in tropical and subtropical intertidal estuarine zones, is a diverse group of salt-tolerant plants and represents a rich resource of fungal endophytes [1,2]. Endophytic fungi are distributed in the tissues of every plant to establish a remarkable mutualistic association with their host plants [3]. As a consequence of this unusual lifestyle, endophytic fungi are widely considered to be a pivotal and prolific reservoir of structurally unique and biologically active secondary metabolites with promising medicinal, agricultural, or industrial applications [4]. The genus *Aspergillus* was first described by Micheli in 1729 and consists of over 300 species distributed worldwide that are able to grow and sporulate under very different environmental conditions including significant changes of temperature, osmolarity, or pH [5,6]. Moreover, *Aspergillus* species are well known for their prolific production of chemically versatile bioactive secondary metabolites [7], such as the potent cholesterol-lowering agent lovastatin obtained from the culture extracts of *Aspergillus terreus* [8].

In the past decade, our research group has focused on mangrove endophytic fungi from the South China Sea to discover new and bioactive metabolites [9,10,11,12]. As part of our ongoing research, a fungal strain *Aspergillus* sp. ZJ-68 was collected from fresh leaves of the mangrove plant *Kandelia candel*. Further chemical investigations of its crude extract led to isolation of five new compounds, including three new benzofuranoids, namely asperfuranoids A–C (**1**–**3**) and two new phenylpropanoid derivatives (**6** and **7**), together with nine known analogues (**4**, **5**, and **8**–**14**) (Figure 1). Compounds **4** and **5** were a pair of enantiomers and the determination of their absolute configurations is an important issue during structure determination. In this context, we report for the first time the absolute configurations of their central chirality elements. In the bioactivity assays, the *α*-glucosidase inhibitory activity and antibacterial activities against four pathogenic bacteria (*S. aureus*, *E. coli*, *B. subtilis*, and *P. aeruginosa*) of **1**−**14** were assessed. Herein, details of the isolation, structure elucidation, and biological activities of these compounds are described.

## 2. Results

### 2.1. Structure Elucidation

The crude extract of *Aspergillus* sp. ZJ-68 was purified by repeatedly various chromatography to yield five new (**1**–**3**, **6**, and **7**) and nine known compounds (**4**, **5**, and **8**−**14**). The known compounds were identical to (−)-penicisochroman A (**4**) [13,14,15], (+)-penicisochroman A (**5**) [13,14,15], 2-(hydroxymethyl)-3-propylphenol (**8**) [16], peniciphenol (**9**) [13], penicibenzoxepinol (**10**) [17], (−)-brassicadiol (**11**) [14,18], (+)-pseudodeflectusin (**12**) [14,19], (−)-penicisochroman B (**13**) [13,14], and ustusorane A (**14**) [14,20] by the comparison of their spectral data (NMR and MS) as well as specific rotation data with those reported.

Asperfuranoid A (**1**) was obtained as a pale yellow oil with a molecular formula C_15_H_22_O_4_, as inferred from the HRESIMS [M + Na]^+^ ion at *m*/*z* 289.14070 (calcd for C_15_H_22_NaO_4_, 289.14103) and NMR data, suggesting five degrees of unsaturation. Examination of the ^1^H NMR data (Table 1) of **1** revealed unambiguously the presence of a 1,2,3,4-tetrasubstituted benzene [δ_H_ 7.04 and 6.87 (each 1H, d, *J* = 7.6 Hz)], one oxygenated methylene [δ_H_ 4.74 and 4.63 (each 1H, d, *J* = 7.6 Hz)], two oxygenated methines [δ_H_ 4.74 (m) and 4.57 (t, *J* = 9.0 Hz)], two methylenes [δ_H_ 3.19 (dd, *J* = 15.7, 8.5 Hz), 3.08 (dd, *J* = 15.7, 9.0 Hz), 1.87 (m), and 1.78 (m)], and three aliphatic methyls [δ_H_ 1.36 (s), 1.14 (s), and 0.94 (t, *J* = 7.4 Hz)]. The ^13^C NMR data (Table 1) of **1** disclosed 15 carbon resonances that were identified by HSQC (Appendix A) as five nonprotonated carbons (four olefinic carbons and one sp^3^ oxygen-bearing carbon), four methine carbons (two olefininc carbons and two sp^3^ oxygen-bearing carbons), three sp^3^ methylene carbons, and three methyl carbons. The molecular formula of **1** required five degrees of unsaturation, but only six olefinic carbons resonating at δ_C_ 158.6 (C, C-9b), 126.6 (C, C-3a), 124.7 (CH, C-4), 118.5 (CH, C-5), 142.5 (C, C-5a), and 119.8 (C, C-9a) were detected, indicating the bicyclic nature of **1**. The planar structure of **1** was further established by the 2D NMR data (Figure 2). A hydrogenated benzofuran ring was deduced by the COSY correlations (H-2/H_2_-3 and H-4/H-5) and the HMBC correlations from H-2 to C-9b (δ_C_ 158.6) and C-3a (δ_C_ 126.6), from H-3 to C-3a, C-4 (δ_C_ 124.7), and C-9b (δ_C_ 158.6), and from H-4 to C-5a (δ_C_ 142.5) and C-9b. The locations of two hydroxy groups at C-6 and C-11 were suggested by the chemical shifts of C-6 (δ_C_ 72.6, δ_H_ 4.74) and C-11 (δ_C_ 71.7). The additional COSY correlations of H-6/H_2_-7/H_3_-10 and the HMBC correlations from H-6 (δ_H_ 4.74) to C-5 (δ_C_ 118.5), C-5a, and C-9a, and from H_2_-7 to C-5a allowed the establishment of a propan-1-ol group, which was linked to C-5a of the benzofuran moiety. The presence of a hydroxyisopropyl group which was assigned at C-2 was deduced from the HMBC correlations from H_3_-12 and H_3_-13 to C-11 (δ_C_ 71.7) and C-2 (δ_C_ 89.6). Finally, the planar structure of **1** was established by connecting the hydroxymethyl group to C-9a as evidenced by the HMBC correlations from H_2_-9 to C-5a, C-9a, and C-9b (Figure 2).

The absolute configuration of the C-6 was determined by the modified Mosher’s method [21]. The △δ*^SR^* values between **1a** and **1b** (*S*-and *R*-MTPA esters of **1** on 6-OH) were negative for H-4/H-5 and positive for H_2_-7/H_3_-10, which indicated the 6*R* configuration (Figure 3). Then the other stand-alone stereogenic center C-2 of **1** was deduced and confirmed by comparison of its experimental and theoretical electronic circular dichroism (ECD) spectra, the latter calculated at the b3lyp/6-311+G (d,p) level (Figure 4). On the basis of the matching of the experimental and computed ECD spectra, the absolute configuration of **1** was defined as 2*R*, 6*R* (Figure 1).

The molecular formula of asperfuranoid B (**2**) was established as C_15_H_20_O_4_ on the basis of the ion peak at *m*/*z* 263.12854 [M − H]^−^ in the HRESIMS spectrum, accounting for six degrees of unsaturation. The ^1^H NMR data (Table 1) of **2** were similar to those of compound **1**, except for the presence of a terminal olefinic methylene [δ_H_ 5.39 (d, *J* = 9.1 Hz), and 5.27 (d, *J* = 10.5 Hz)] and an olefinic proton resonating at δ_H_ 6.13 (H-7), and the absence of two aliphatic signals at δ_H_ 1.87 and 1.78 (H_2_-7), and δ_H_ 0.94 (H_3_-10), suggesting a double bond at △^7(10)^. This deduction was further corroborated by the cross-peaks of H_2_-10/H-7/H-6 in the COSY spectrum and the HMBC correlations (Figure 2) from H-6 to C-7 (δ_C_ 139.3) and C-10 (δ_C_ 115.3), and from H-7 to C-5a (δ_C_ 141.0). In addition, the CD curve and specific rotation of **2** was similar to the CD spectrum and specific rotation of **1** in MeOH (Figure 4), respectively. Therefore, the absolute configuration of compound **2** was also determined as 2*R*, 6*R*.

Asperfuranoid C (**3**) has a molecular formula of C_15_H_16_O_4_, as determined by the HRESIMS and NMR data, requiring eight degrees of unsaturation. The 2D NMR data (Figure 2) provided the structure of **3** to be partially related to **1**, possessing a benzofuran ring with the substitution of hydroxyisopropyl at C-2. These functionalities accounted for six sites of unsaturation. The remaining NMR resonances (Table 1) were attributed to two olefinic carbons (δ_C_ 101.7 and 160.7) for a double bond, a ketone group (δ_C_ 199.9), and a methyl group (δ_C_ 20.1), providing two additional sites of unsaturation. The HMBC correlations from the hydroxymethyl protons (H_2_-9) at δ_H_ 5.28 and 5.22 to C-5a (δ_C_ 142.6), C-7 (δ_C_ 160.7), C-9a (δ_C_ 108.9), and C-9b (δ_C_ 167.9), from the olefinic proton H-6 (δ_H_ 5.69) to C-5 (δ_C_ 117.6), C-5a, and C-9a, and from the methyl protons at δ_H_ 1.98 (s, H_3_-10) to C-6 (δ_C_ 101.7), and C-7 (Figure 2) allowed the establishment of a 7-methyl-9H-pyran ring between C-5a and C-9a, accounting for the remaining one site of unsaturation. Additionally, the HMBC correlations from the H-2 and H-4 to C-3 (δ_C_ 199.9) (Figure 2) confirmed the presence of the ketone group, which was located at C-3a. Thus, the planar structure of **3** was confirmed and its absolute configuration was designated as 2*S* by experimental and theoretical ECD data (Figure 4).

Compounds **4** and **5** were obtained as pale yellow powders with the molecular formula of C_16_H_18_O_4_ based on HRESIMS data. Analysis of the NMR data (Table 2) led to the identification of **4** and **5** as penicisochroman A, a previously reported and structurally characterized compound isolated from *Penicillium* sp. [13]. The near zero optical rotation, congruent with the crystal data (centrosymmetric space group *P*$\bar{1})$) which have been reported from our research [15], indicated a racemic mixture. The chiral HPLC separation of (±)-penicisochroman A was further performed on an Acchrom *S*-Chiral A column to yield a pair of enantiomers, (−)-penicisochroman A (**4**) and (+)-penicisochroman A (**5**), with the opposite Cotton effects and the opposite optical rotations. To determine the absolute configurations of (−)-**4** and (+)-**5**, the ECD spectra of (−)-**4** and (+)-**5** were measured in MeOH and compared with the calculated ECD spectra of the enatiomers (Figure 5). Thus, the absolute configurations of (−)-**4** and (+)-**5** were determined as 7*R* and 7*S* (Figure 1), respectively.

Asperpanoid A (**6**) had a molecular formula of C_10_H_14_O_3_, as determined by the HRESIMS and NMR data, requiring four degrees of unsaturation. The ^1^H NMR data (Table 3) and HSQC spectrum (Appendix A) provided the resonances for a methyl group [δ_H_ 0.91 (3H, t, *J* = 7.3 Hz)], two methylene signals [δ_H_ 2.50 (2H, m) and 1.51 (2H, m)], a hydroxymethyl group [δ_H_ 4.83 (2H, s)], and a 1,2,3,4-tetrasubstituted benzene [δ_H_ 6.66 and 6.51 (each 1H, d, *J* = 8.1 Hz)]. Analyses of the 1D and 2D NMR data (Figure 2) established the gross structure of **6** to be similar as a known phenylpropanoid analogue, 2-(hydroxymethyl)-3-propylphenol (**8**) [16]. The signals at δ_H_ 6.66 (d, *J* = 8.1 Hz) and 6.51 (d, *J* = 8.1 Hz) indicated the presence of a 1,2,3,4-tetrasubstituted benzene system in **6**, instead of the trisubstituted benzene in **8**. The ^13^C NMR spectrum (Table 3) revealed that a deshielded aromatic nonprotonated carbon (δ_C_ 144.1, C-1) in **6** replaced an aromatic methine carbon (δ_C_ 114.4, C-1) in **8**. These observations coupled with the MS data suggested that the position of the additional hydroxy group was located at C-1. The COSY correlation between H-2 and H-3, and the HMBC correlations from H-3 to C-1 and C-5 confirmed the above deductions. Thus, compound **6** was the 1-hydroxylated analogue of **8** (Figure 1).

Asperpanoid B (**7**) was assigned the molecular formula C_11_H_12_O_3_ on the basis of HRESIMS data. The ^1^H and ^13^C NMR data (Table 3) of **7** matched well with those for the known compound peniciphenol (**9**) and indicated the same structural features present in **9** except for the presence of methoxy group at C-6, which is consistent with the difference in the molecular formula [13]. Accordingly, the signal for the additional methoxy group at δ_H_ 3.85 and δ_C_ 56.1 were observed in the NMR spectra of **7**. These observations coupled with the MS data indicate that the hydroxyl group at C-6 in **9** was replaced by a methoxy group in **7**. The location of the methoxy group at C-6 was further confirmed by the HMBC correlation from the methoxy proton to C-6. The similar NOESY coupled patterns (Figure 2) and coupling constant between H-7 and H-8 (*J*_7,8_ = 11.5 Hz) to **9** assigned the same Z-Δ^7^-double bond in **7**. Therefore, compound **7** was the methoxy derivative of **9** (Figure 1).

### 2.2. Biological Activity

The isolated compounds **1**–**14** were evaluated for their inhibitory activity against *α*-glucosidase in vitro, and acarbose was selected as the positive control (IC_50_ = 453.3 μM) (Table 4). Compounds exhibiting inhibitory activity against *α*-glucosidase with values greater than 40% at 100 μM were further tested and IC_50_ values calculated. Compound **6** was more potent than acarbose, with an IC_50_ value of 12.4 μM. Additionally, the other compounds revealed weak or no inhibitory effects at a concentration of 100 μM.

All isolated compounds (**1**−**14**) were also tested for their antibacterial effects against four pathogenic bacteria (*S. aureus*, *E. coli*, *B. subtilis*, and *P. aeruginosa*) (Table 5). Compounds **8** and **11** showed activities against *S. aureus*, *E. coli*, and *B. subtilis*, with MIC values in the range of 4.15 to 12.5 μg/mL, while the other compounds exhibited weak or no antibacterial activities (MIC values > 100 μg/mL). None of the compounds were active against *P. aeruginosa* (MIC values > 100 μg/mL).

## 3. Materials and Methods 

### 3.1. General Experimental Procedures

The melting points were recorded on a SGW X-4B micro melting point apparatus (Shanghai Precision Scientific Instrument Co., Ltd, Shanghai, China) and were uncorrected. Optical rotations were determined using an MCP 300 polarimeter (Anton Paar, Graz, Austria) at 25 °C. UV data were recorded on a TU-1900 spectrophotometer (Persee, Beijing, China) in MeOH solution. ECD spectra were recorded using a Chirascan CD spectrometer (Applied Photophysics, London, UK). IR spectra were recorded on a Nicolet Nexus 670 spectrophotometer (Thermo Nicolet Corporation, Madison, WI, USA) in KBr discs. All NMR experiments were measured with Bruker Avance 500 spectrometers (500 and 125 MHz) (Bruker BioSpin, Switzerland), and the residual solvent peaks of CDCl_3_ (δ_C_ 77.1/δ_H_ 7.26), acetone-*d*_6_ (δ_C_ 29.8 and 206.3/δ_H_ 2.05), or methanol-*d*_4_ (δ_C_ 49.0/δ_H_ 3.31) were used as references. HRESIMS data were acquired on a Thermo Fisher LTQ Orbitrap Elite high-resolution mass spectrometer (Thermo Fisher Scientific, Waltham, MA, USA). Column chromatography (CC) was carried by silica gel (200–300 mesh, Qingdao Maine Chemical Factory) and Sephadex LH-20 (GE Healthcare Bio-Sciences AB, Stockholm, Sweden). Semipreparative HPLC was performed on a Primaide HPLC system (Hitachi Instrument Dalian Co., Ltd, Dalian, China) with an Ultimate XB-C18 column (10 × 250 mm, 10 μm). 

### 3.2. Fungal Material

The fungus *Aspergillus* sp. ZJ-68 (Appendix A) was isolated from fresh leaves of the mangrove plant *Kandelia candel*, which were collected in July 2016 from the Zhanjiang Mangrove Nature Reserve in Guangdong Province, China. The fungal strain was identified according to a molecular biology protocol by rDNA amplification and sequencing of the internal transcribed spacer (ITS) region [22]. A BLAST search result showed that it was most similar (99%) to the sequence of *Aspergillus* sp. (compared to JF312217.1). The sequence data have been submitted to GenBank with accession number MK629267. The isolate was stored on PDA medium (potato 200 g, dextrose 20 g, sea salt 3 g, and agar 15 g in 1.0 L of H_2_O, pH 7.4–7.8) slants at 4 °C.

### 3.3. Fermentation

The fungus *Aspergillus* sp. ZJ-68 was cultured on PDA agar plate at 28 °C for 7 days. The fungal colony was further inoculated into the PDB medium (potato 200 g, dextrose 10 g, and sea salt 3 g in 1.0 L of H_2_O, pH 7.4–7.8) at 28 °C for 3 days on a rotating shaker (140 rpm). Then, a large-scale fermentation of the strain was performed. The fungal seed broth (10 mL) was added to one hundred flasks (1000 mL), each containing 300 mL of liquid medium that was composed of potato 200 g, dextrose 20 g, and sea salt 30 g in 1.0 L of H_2_O, pH 7.4–7.8. These flasks were incubated at 28 °C for 30 days under static conditions. 

### 3.4. Extraction and Isolation

The whole fermentation broth (30 L) was filtered by cheesecloth to separate the mycelia from the filtrate. The mycelia were extracted three times by CH_2_Cl_2_, while the filtrate was extracted three times by the equivalent volume of EtOAc. The CH_2_Cl_2_ and EtOAc solutions were concentrated under reduced pressure to give an organic extract. This extract was fractionated by silica gel (200–300 mesh) column chromatography using a petroleum ether (PE, 60–90 °C) and EtOAc gradient system (from 1:0 to 0:1) to give 10 fractions (F1 to F10). Fraction F2 was applied to Sephadex LH-20 eluting with CH_2_Cl_2_–MeOH (1:1, *v*/*v*) and purified by semipreparative reversed-phase (RP) HPLC column (80% MeOH/H_2_O) to afford the mixture containing **4** and **5** (10.1 mg, t*_R_*: 20.6 min), **12** (31.4 mg, t*_R_*: 18.0 min), and **13** (23.4 mg, t*_R_*: 19.1 min). The chiral resolutions of **4** (3.6 mg) and **5** (4.1 mg) were performed on an Acchrom *S*-Chiral A column (10 × 250 mm, 5 μm) using hexanes–isopropyl alcohol (90:10, *v*/*v*) as the eluent (Appendix A). Fraction F3 was separated into two subfractions (F3a and F3b) by CC on silica gel eluting with a step gradient of PE/EtOAc (80:20 to 70:30, *v*/*v*). Compounds **3** (3.4 mg, t*_R_*: 20.1 min), **7** (6.8 mg, t*_R_*: 19.6 min), and **10** (8.9 mg, t*_R_*: 18.0 min) were obtained from the F3a which was subjected on RP-HPLC column (70% MeOH–H_2_O). F3b was submitted to silica gel CC (CH_2_Cl_2_–MeOH, 2:100, *v*/*v*) to yield **9** (4.5 mg), **11** (9.8 mg), and **14** (7.4 mg). Fraction F4 was fractionated by CC on silica gel into two subfractions (F4a and F4b) eluting with gradient CH_2_Cl_2_ and MeOH (2:100 to 3:100, *v*/*v*). Compounds **1** (10.8 mg, t*_R_*: 19.7 min) and **2** (5.3 mg, t*_R_*: 20.8 min) were purified by RP-HPLC column (70% MeOH–H_2_O) from the F4b. F4a was submitted to silica gel CC eluting with CH_2_Cl_2_–MeOH (3:100, *v*/*v*) and further purified by Sephadex LH-20 eluting with CH_2_Cl_2_–MeOH (1:1, *v*/*v*) to yield compounds **6** (6.8 mg) and **8** (5.9 mg).

Asperfuranoid A (**1**): Pale yellow oil; [*α*]${}_{D}^{25}$ −34.8 (c 0.03, MeOH); UV (MeOH) *λ*_max_ (log *ɛ*) 205 (3.68), 290 (1.02) nm; ECD (0.15 mM, MeOH) *λ*_max_ (∆*ɛ*) 206 (−6.59), 234 (−2.11), 260 (−0.54), and 288 (+0.49) nm; IR (KBr) *ν*_max_ 3340, 2956, 2867, 1672, 1592, 1440, 1378, 1193, 971 cm^−1^; ^1^H and ^13^C NMR data, Table 1; HRESIMS *m*/*z* 289.14070 [M + Na]^+^ (calcd for C_15_H_22_NaO_4_, 289.14103).

Asperfuranoid B (**2**): Colorless oil; [*α*]${}_{D}^{25}$ −13.5 (c 0.02, MeOH); UV (MeOH) *λ*_max_ (log *ɛ*) 206 (3.53), 291 (1.05) nm; ECD (0.05 mM, MeOH) *λ*_max_ (∆*ɛ*) 202 (+1.83), 213 (−3.87), 237 (−1.63), 265 (−0.73), and 288 (+0.51) nm; IR (KBr) *ν*_max_ 3369, 2919, 1764, 1712, 1643, 1592, 1446, 1253, 991 cm^−1^; ^1^H and ^13^C NMR data, Table 1; HRESIMS *m*/*z* 263.12854 [M − H] ^−^ (calcd for C_15_H_19_O_4_, 263.12888).

Asperfuranoid C (**3**): Pale yellow oil; [*α*]${}_{D}^{25}$ −99 (c 0.5 MeOH); UV (MeOH) *λ*_max_ (log *ɛ*) 215 (3.31), 280 (2.01), 352 (1.02) nm; ECD (0.14 mM, MeOH) *λ*_max_ (∆*ɛ*) 216 (−21.01), 227 (+3.01), 313 (+2.02), and 343 (−0.96) nm; IR (KBr) ν_max_ 3423, 2927, 1710, 1604, 1440, 1376, 1274, 1170, 1072 cm^−1^; ^1^H and ^13^C NMR data, Table 1; HRESIMS *m*/*z* 259.09756 [M − H]^−^ (calcd for C_15_H_15_O_4_, 259.09758).

(−)-Penicisochroman A (**4**): Pale yellow powder; mp 171–172 °C; [*α*]${}_{D}^{25}$ −111 (c 0.5 MeOH); UV (MeOH) *λ*_max_ (log *ɛ*) 220 (3.23), 286 (2.98), 351 (1.02) nm; ECD (0.11 mM, MeOH) *λ*_max_ (∆*ɛ*) 231 (+2.34), 282 (+1.71), and 348 (−0.35) nm; IR (KBr) ν_max_ 3417, 2921, 2850, 1712, 1602, 1434, 1375, 1265, 1093 cm^−1^; ^1^H and ^13^C NMR data, Table 2; HRESIMS *m*/*z* 275.12767 [M + H]^+^ (calcd for C_16_H_19_O_4_, 275.12779).

(+)-Penicisochroman A (**5**): Pale yellow powder; mp 170–171 °C; [*α*]${}_{D}^{25}$ +105 (c 0.5 MeOH); UV (MeOH) *λ*_max_ (log *ɛ*) 220 (3.23), 286 (2.98), 351 (1.02) nm; ECD (0.11 mM, MeOH) *λ*_max_ (∆ɛ) 233 (−2.62), 291 (−1.70), and 343 (+0.16) nm; IR (KBr) ν_max_ 3403, 2927, 2856, 1712, 1610, 1436, 1376, 1268, 1085 cm^−1^; ^1^H and ^13^C NMR data, Table 2; HRESIMS *m*/*z* 275.12770 [M + H]^+^ (calcd for C_16_H_19_O_4_, 275.12779).

Asperpanoid A (**6**): Colorless oil; UV (MeOH) *λ*_max_ (log *ɛ*) 224 (2.36), 281 (2.01) nm; IR (KBr) ν_max_ 3340, 2956, 2925, 2867, 1585, 1461, 1253, 987, 786 cm^−1^; ^1^H and ^13^C NMR data, Table 3; HRESIMS *m*/*z* 181.08690 [M − H]^−^ (calcd for C_10_H_13_O_3_, 181.08702).

Asperpanoid B (**7**): Colorless powder; mp 116–117 °C; UV (MeOH) *λ*_max_ (log *ɛ*) 213 (3.35), 243 (249), 284 (2.19) nm; IR (KBr) *ν*_max_ 3232, 2967, 2917, 1685, 1573, 1456, 1265, 1074, 1000, 717 cm^−1^; ^1^H and ^13^C NMR data, Table 3; HRESIMS *m*/*z* 217.08336 [M + Na]^+^ (calcd for C_11_H_14_NaO_3_, 217.08352).

### 3.5. Inhibitory Activity Against α-Glucosidase

*α*-Glucosidase inhibitory activity was assessed in 96-well plates using 0.01 M KH_2_PO_4_−K_2_HPO_4_ (pH 7.0) buffer solution, and the enzyme solutions were prepared to give 2.0 units/mL in buffers. The assay was conducted in the 200 μL reaction system which contains 168 μL of buffers, 10 μL of diluted enzyme solution, and 2 μL of dimethyl sulfoxide (DMSO) or sample (dissolved in DMSO). The detailed methodology for biological testing has already been described in a previous report [12]. Acarbose was used as the positive control and all assays were performed in three replicates.

### 3.6. Antibacterial Assay 

Two Gram-positive bacteria *B. subtilis* (ATCC 6633) and *S. aureus* (ATCC 25923), and two Gram-negative bacteria *E. coli* (ATCC 25922) and *P. aeruginosa* (ATCC 27853) were used. The antibacterial assay and the determination of the MIC were assessed according to continuous dilution method in 96-well plates which has been described in our published paper [23]. The tested substances were dissolved in DMSO and ciprofloxacin was the positive control.

## 4. Conclusions

In summary, three new benzofuranoids, asperfuranoids A–C (**1**–**3**), and two new phenylpropanoid derivatives (**6** and **7**) were isolated from the mangrove endopytic fungus *Aspergillus* sp. ZJ-68. The absolute configurations of **1**–**5** were unambiguously established by a combination of Mosher’s method, and experimental and calculated ECD data. Compounds **4** and **5** were a pair of enantiomers and their absolute configurations were established as 7*R* and 7*S* for the first time. This study further expanded the structural diversity of naturally occurring benzofuranoid and phenylpropanoid derivatives. In the bioactivity assays, compound **6** exhibited potent inhibitory activity against *α*-glucosidase with an IC_50_ value of 12.4 μM, and compounds **8** and **11** showed significant activities against *S. aureus*, *E. coli*, and *B. subtilis* with MIC values in the range of 4.15 to 12.5 μg/mL. Compounds **6**, **8**, and **11** may be considered as potential new drug leads.

## Figures and Tables

**Figure 1 marinedrugs-17-00478-f001:**
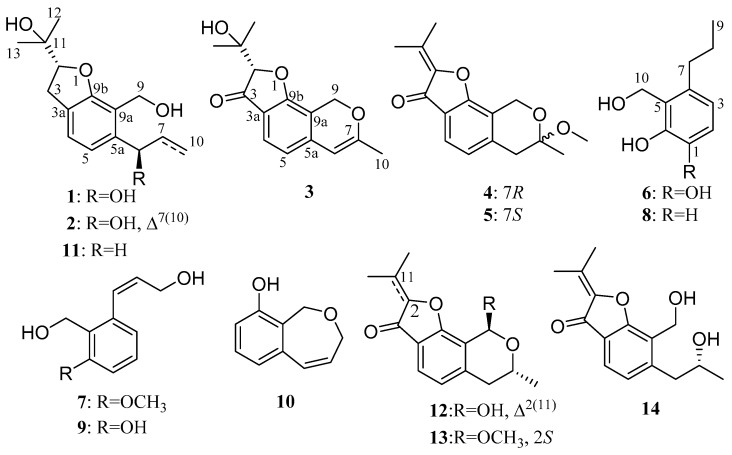
Structures of compounds **1**–**14**.

**Figure 2 marinedrugs-17-00478-f002:**
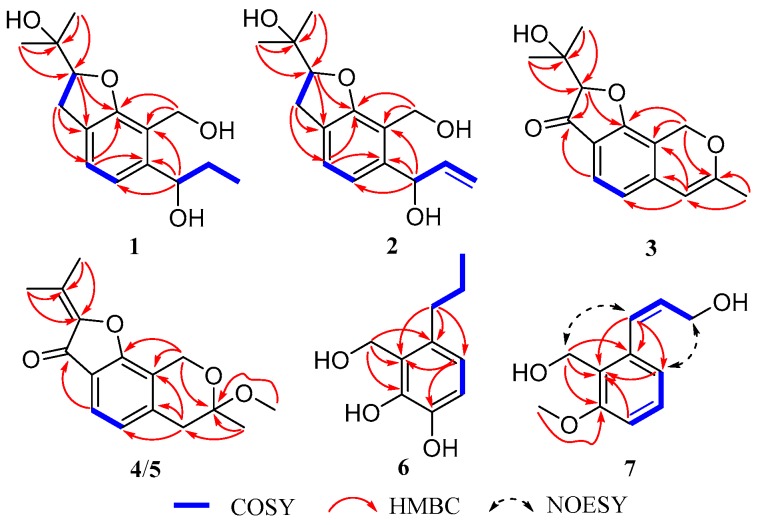
Key 2D NMR correlations for **1**–**7**.

**Figure 3 marinedrugs-17-00478-f003:**
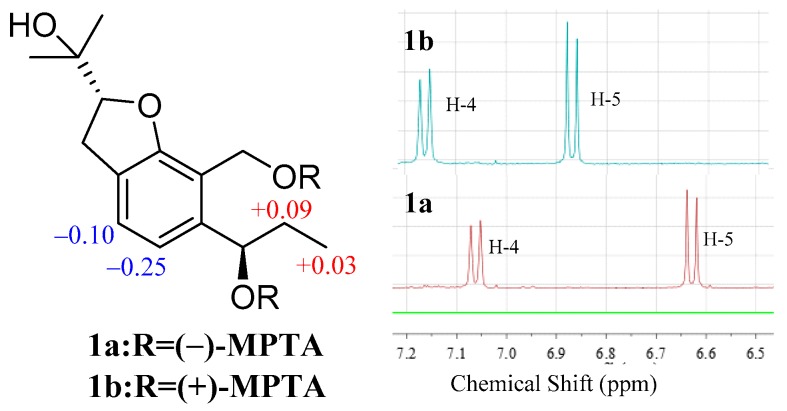
∆δ (= δ*_S_* − δ*_R_*) values for (*S*)- and (*R*)-MTPA esters of **1**.

**Figure 4 marinedrugs-17-00478-f004:**
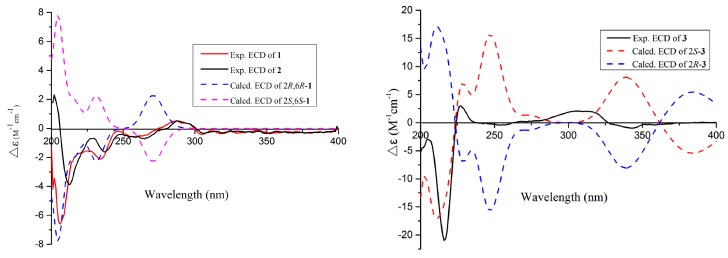
Experimental electronic circular dichroism (ECD) spectra of **1**−**3** in MeOH and the calculated ECD spectra of **1** and **3** at the B3LYP/6-311 + G (d, p) level.

**Figure 5 marinedrugs-17-00478-f005:**
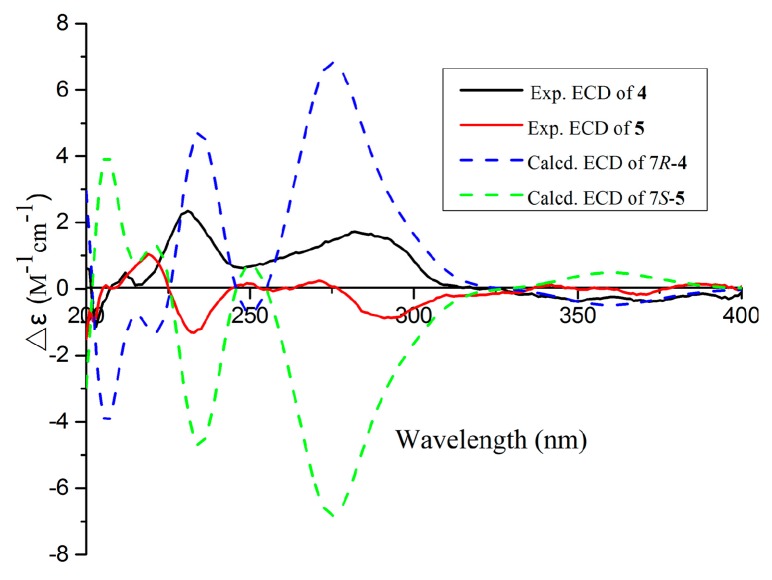
Experimental and calculated ECD spectra of **4** and **5**.

**Table 1 marinedrugs-17-00478-t001:** ^1^H (500 MHz) and ^13^C (125 MHz) NMR data of compounds **1**–**3** in CDCl_3_ (δ in ppm).

No.	1		2		3	
	δ_C_, Type	δ_H_, mult (*J*/Hz)	δ_C_, Type	δ_H_, mult (*J*/Hz)	δ_C_, Type	δ_H_, mult (*J*/Hz)
2	89.6, CH	4.57, t (9.0)	89.6, CH	4.60, t (8.9)	89.7, CH	4.37, s
3	30.7, CH_2_	3.19, dd (8.5, 15.7) 3.08, dd (9.5, 15.6)	30.7, CH_2_	3.21, dd (8.4, 15.8) 3.11, dd (9.5, 15.8)	199.9, C	
3a	126.6, C		127.5, C		119.7, C	
4	124.7, CH	7.04, d (7.6)	124.6, CH	7.06, d (7.6)	124.2, CH	7.47, d (7.9)
5	118.5, CH	6.87, d (7.6)	120.0, CH	6.84, d (7.6)	117.6, CH	6.63, d (7.9)
5a	142.5, C		141.0, C		142.6, C	
6	72.6, CH	4.74, m	72.6, CH	5.41, m	101.7, CH	5.69, s
7	30.2, CH_2_	1.87, m 1.78, m	139.3, CH	6.13, ddd (4.6, 10.5, 17.5)	160.7, C	
9	55.9, CH_2_	4.74, d (12.0) 4.63, d (12.0)	56.2, CH_2_	4.81, d (12.1) 4.69, d (12.1)	62.7, CH_2_	5.28, d (13.2) 5.22, d (13.2)
9a	119.8, C		120.2, C		108.9, C	
9b	158.6, C		158.9, C		167.9, C	
10	10.9, CH_3_	0.94, t (7.4)	115.3, CH_2_	5.39, d (9.1) 5.27, d (10.5)	20.1, CH_3_	1.98, s
11	71.7, C		71.8, C		72.6, C	
12	24.3, CH_3_	1.14, s	24.2, CH_3_	1.17, s	24.1, CH_3_	1.20, s
13	26.7, CH_3_	1.36, s	26.6, CH_3_	1.36, s	26.2, CH_3_	1.36, s

**Table 2 marinedrugs-17-00478-t002:** ^1^H (500 MHz) and ^13^C (125 MHz) NMR data of compounds **4** and **5** in CDCl_3_ (δ in ppm).

No.	4		5	
	δ_C_, Type	δ_H_, mult (*J*/Hz)	δ_C_, Type	δ_H_, mult (*J*/Hz)
2	145.5, C		145.5, C	
3	183.6, C		183.6, C	
3a	121.3, C		121.3, C	
4	122.1, CH	7.53, d (7.9)	122.1, CH	7.53, d (7.9)
5	123.2, CH	6.84, d (7.9)	123.2, CH	6.84, d (7.9)
5a	140.8, C		140.9, C	
6	39.5, CH_2_	3.00, d (16.2) 2.92, d (16.2)	39.4, CH_2_	3.00, d (16.5) 2.92, d (16.5)
7	97.6, C		97.6, C	
9	57.9, CH_2_	4.93, d (15.6) 4.72, d (15.6)	57.9, CH_2_	4.93, d (15.6) 4.72, d (15.6)
9a	118.4, C		118.4, C	
9b	160.5, C		160.5, C	
10	23.1, CH_3_	1.53, s	23.1, CH_3_	1.53, s
11	131.8, C		131.8, C	
12	17.6, CH_3_	2.36, s	17.6, CH_3_	2.36, s
13	20.3, CH_3_	2.09, s	20.3, CH_3_	2.09, s
7-OCH_3_	49.2, CH_3_	3.34, s	49.2, CH_3_	3.34, s

**Table 3 marinedrugs-17-00478-t003:** NMR spectroscopic data for compounds **6** and **7** (δ in ppm).

No	6*^a^*		7*^b^*	
	δ_C_, Type	δ_H_, mult (*J*/Hz)	δ_C_, Type	δ_H_, mult (*J*/Hz)
1	144.1, C		111.0, CH	6.93, d (8.3)
2	114.7, CH	6.66, d (8.1)	129.7, CH	7.25, t (7.9)
3	121.1, CH	6.51, d (8.1)	122.8, CH	6.74, d (7.6)
4	132.6, C		139.0, C	
5	125.2, C		127.7, C	
6	145.3, C		159.6, C	
7	35.2, CH_2_	2.50, m	130.2, CH	6.79, d (11.5)
8	25.6, CH_2_	1.51, m	133.4, CH	5.94, dt (6.7, 11.5)
9	14.2, CH_3_	0.91, t (7.3)	59.5, CH_2_	4.10, dd (1.3, 6.8)
10	59.2, CH_2_	4.83, s	56.6, CH_2_	4.64, s
6-OCH_3_			56.1, CH_3_	3.85, s

**Table 4 marinedrugs-17-00478-t004:** Inhibitory effects of compounds **1**−**14** against *α*-glucosidase.

Compounds *^a^*	% Inhibition (100 μM)	IC_50_ (μM)
**6**	98	12.4 ± 1.0
Acarbose *^b^*	19	453.3 ± 1.0

*^a^* Compounds **1**−**5** and **7**−**14** showed weak or no activity (IC_50_ > 100 μM); *^b^* acarbose was tested as positive control.

**Table 5 marinedrugs-17-00478-t005:** Antibacterial activities of compounds **1**–**14**.

Compounds *^a^*	MIC (μg/mL)			
	*S. aureus*	*E. coli*	*B. subtilis*	*P. aeruginosa*
**8**	4.15 ± 1.12	8.3 ± 1.0	8.3 ± 1.1	>100
**11**	12.5 ± 1.1	12.5 ± 1.2	12.5 ± 1.0	>100
ciprofloxacin*^b^*	1.25 ± 1.10	1.25 ± 1.12	2.5 ± 1.1	2.5 ± 1.2

*^a^* Compounds **1**–**7**, **9**, **10**, and **12**–**14** showed no activity (MIC > 100 μg/mL); *^b^* ciprofloxacin was tested as positive control.

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
