# Peer review of "New Benzofuranoids and Phenylpropanoids from the Mangrove Endophytic Fungus, Aspergillus sp. ZJ-68"

_marinedrugs, 2019, doi:10.3390/md17080478_

Round 1

Reviewer 1 Report

The paper by Li, She, and co-workers details their isolation of natural products from the culture of Aspergillus sp. ZJ-68. Isolation was carried out by a combination of extraction and chromatography, and the structures were elucidated by 1 and 2 dimensional NMR. Absolute configuration was guided by ECD. Activity of the natural products was determined by testing them against a glucosidase in vitro, and antibacterial activity was measured against both Gram-positive and Gram-negative bacteria.

There are some significant issues with the interpretation of the data. The authors claim that compounds 4 and 5 are new, yet they are already in the literature (penicisochromatin A) having been isolated in 2016 from Aspergillus (Beilstein J. Org. Chem), and was also isolated in 2010 in reference number 14 of this manuscript.

Furthermore, this reviewer has concerns about whether compounds 4 and 5 in this manuscript are even natural products. We know from compound 3 that enol ethers are made by this fungus. The author’s protocol for isolation of these compounds involves the use of chromatographic methods that use methanol. Although it is not specifically mentioned in the protocol, most reverse phase hplc methods use some sort of acid, and in this case the acid could be making the new compounds via methanolysis of the enol ether.

For this paper to be published, compounds 4 and 5 will need to be presented as known compounds. The compounds names (page 8 and 9) will need to be changed to account for this.

Minor errors:

Phenypropanoid vs phenylpropanoid --- The title uses the word phenypropanoid, which seemingly is an error used throughout the paper.

Page 1, line 19 – an “and” is required after “Mosher’s method,”

Page 1, line 23 – “exhibited potent”, not “exhibited a potent”

Page 1, line 29 – “The mangrove forest”, not “Mangrove forest”

Page 1, line 30 – “salt-tolerant plants”, not “salt-tolerant plant”

Page 1, line 31 – “the tissues”, not “tissues”

Page 1, line 43 – “from fresh leaves”, not “from a fresh leaves”

Page 6, line 155 – “methine carbon”, not “methine ccarbon”

Page 6, line 159 – “Figure 1”, not “Figure1”

Page 6, line 171 – “methoxy derivative”, not “methyl derivative”

Page 7, line 176 – “IC50 values calculated”, not “calculated IC50 values”

Page 7, line 201 – “from fresh leaves”, not “from the fresh leaves”

Page 7, line 203 – “biology”, not “biological”

Page 8, line 204 – needs reference after “region.”

Page 9, line 264 – “in 96-well plates”, not “in the 96-well plates”

Page 9, line 271 – “ATCC 6633”, not “ATCC6633”

Page 9, line 274 - “in 96-well plates”, not “in the 96-well plates”

Page 9, line 280 – an “and” is required after “Mosher’s method,”

Page 9, line 282 – “exhibited potent”, not “exhibited a potent”

Page 9, line 284/5 – “Those of them may be” should be replaced by naming the compound numbers

Reviewer 2 Report

The manuscript entitled “New Benzofuranoids and Phenypropanoids from the Mangrove Endophytic Fungus, Aspergillus sp. ZJ-68” describes the extraction, isolation, purification and characterization of some new and some reported natural products. Moreover, the isolated compounds were evaluated for their enzyme inhibitory activity against α-glucosidase and antibacterial activities. The experimental work is nicely performed and the manuscript is well written. However, this reviewer has some concerns that should be carefully addressed before accepting this manuscript in Marine Drugs as an article.  

Even though the authors characterize the newly isolated compounds using H NMR, C NMR, and some 2D NMR techniques, X-ray crystallography is the best means of structure elucidation of newly isolated compounds. If possible, the authors should try to crystallize some new solid molecules for example 4, 5 and 7 for X-ray crystallography. The authors should provide melting points of the solid compounds. Even though the compounds 8-14 are reported, the authors should provide at least H NMR, C NMR and HRMS data and spectra of these molecules. For fungus characterization, it would be better to provide rDNA sequencing data in supporting information. Moreover, some simple pictures of the fungus showing colonies of strain and microscopic pictures showing hyphae and conidia of strain will be more informative for readers.

Round 2

Reviewer 2 Report

The authors noted and revised the reviewers comments. Therefore, this reviewer recommends this manuscript for publication in Marine Drugs.